# An Analysis of Early Results after Valve Replacement in Isolated Aortic Valve Stenosis by Using Sutureless vs. Stented Bioprostheses: A Single-Center Middle-Income Country Experience

**DOI:** 10.3390/medicina59061032

**Published:** 2023-05-26

**Authors:** Marko Kaitovic, Slobodan Micovic, Ivan Nesic, Tatjana Raickovic, Jelena Dotlic, Ivan Stojanovic, Tatjana Gazibara

**Affiliations:** 1Cardiac Surgery Department, Institute for Cardiovascular Diseases “Dedinje”, 11040 Belgrade, Serbia; marko.kaitovic@med.bg.ac.rs (M.K.);; 2Department of Surgery, Faculty of Medicine, University of Belgrade, 11000 Belgrade, Serbia; 3Clinic for Obstetrics and Gynecology, Clinical Center of Serbia, 11000 Belgrade, Serbia; 4Department of Gynecology and Obstetrics, Faculty of Medicine, University of Belgrade, 11000 Belgrade, Serbia; 5Institute of Epidemiology, Faculty of Medicine, University of Belgrade, 11000 Belgrade, Serbia

**Keywords:** aortic valve replacement, stented bioprosthesis, sutureless bioprosthesis, Perceval valve, survival, middle-income country

## Abstract

*Background and Objectives*: There is a lack of data about the survival of patients after the implantation of sutureless relative to stented bioprostheses in middle-income settings. The objective of this study was to compare the survival of people with isolated severe aortic stenosis after the implantation of sutureless and stented bioprostheses in a tertiary referral center in Serbia. *Materials and Methods*: This retrospective cohort study included all people treated for isolated severe aortic stenosis with sutureless and stented bioprostheses from 1 January 2018 to 1 July 2021 at the Institute for Cardiovascular Diseases “Dedinje”. Demographic, clinical, perioperative and postoperative data were extracted from the medical records. The follow-up lasted for a median of 2 years. *Results*: The study sample comprised a total of 238 people with a stented (conventional) bioprosthesis and 101 people with a sutureless bioprosthesis (Perceval). Over the follow-up, 13.9% of people who received the conventional and 10.9% of people who received the Perceval valve died (*p* = 0.400). No difference in the overall survival was observed (*p* = 0.797). The multivariate Cox proportional hazard model suggested that being older, having a higher preoperative EuroScore II, having a stroke over the follow-up period and having valve-related complications were independently associated with all-cause mortality over a median of 2 years after the bioprosthesis implantation. *Conclusions*: This research conducted in a middle-income country supports previous findings in high-income countries regarding the survival of people with sutureless and stented valves. Survival after bioprosthesis implantation should be monitored long-term to ensure optimum postoperative outcomes.

## 1. Introduction

Severe aortic stenosis is the most common degenerative valvular heart disease in the industrialized countries [1,2]. Surgical aortic valve replacement is still a standard treatment of low-risk patients with severe aortic stenosis [3,4]. It is estimated that without the surgical intervention, people with symptomatic severe aortic stenosis may expect to live for approximately 3 years [5]. For this reason, technological improvement in the making of biological valves is continually evolving to optimize the surgical management and durability of the bioprostheses and ensure a longer survival and overall better clinical outcomes.

Over the past decade, sutureless valves have been increasingly used in the treatment of severe aortic stenosis. Their relatively easy implantation technique—no need to suture the valve to the aortic annulus—allows the surgeons to cut down the operative time and to apply a minimally invasive surgical approach [6]. Previous studies conducted in the industrialized countries found no differences in one-year mortality rates following the implantation of stented and sutureless bioprostheses [7,8]. However, there seemed to be a better hemodynamic performance in sutureless bioprostheses [8]. Still, the cost of the sutureless bioprostheses is not to be underestimated. Despite similar charges for hospital stay after the implantation of sutureless and stented bioprostheses [9], sutureless bioprostheses can be more than 10 times more expensive compared to the conventional stented bioprostheses. The costs of the sutureless bioprostheses may restrict the access to these valves in settings with limited resources.

The Republic of Serbia is classified as a middle-income country according to the gross domestic product per capita [10]. Its residents have a universal access to health care. Sutureless bioprostheses have been routinely used over the last 5 years, which is somewhat shorter compared to high-income countries. The vast majority of comparative studies of sutureless and stented bioprostheses have been conducted in the industrialized countries [7,8,11]. Yet, there is paucity of evidence about the survival and valve-related complications after the implantation of sutureless relative to stented bioprostheses in middle-income settings. This comparison is relevant because middle-income countries have fewer cardiac surgeons and fewer centers where cardiac surgery is performed [12]. In addition, a shortage of cardiac surgeons leads to the crowding in hospitals and long waiting lists for people needing cardiac surgery [13]. As a result, surgical outcomes may be poorer compared to those observed in high-income countries [14].

Bearing in mind all mentioned above, the aim of this study was to compare the survival of people with severe aortic stenosis after the implantation of sutureless and stented bioprostheses as well as to examine potential predictors of poorer survival after the surgical treatment of severe aortic stenosis in a referral tertiary center in Serbia.

## 2. Materials and Methods

### 2.1. Setting

This research is based on a retrospective cohort study. This study was conducted at the Institute for Cardiovascular Diseases “Dedinje” (Institute). The Institute is a highly specialized tertiary center with more than 200 hospital beds and more than 2000 cardiac surgeries per year. Because of this, it is a regional referral center for all cardiac and vascular-related diseases and a leader in new surgical approaches, such as the micro-bypass (article in press), endoscopic aortic and mitral valve surgery and treatment of end-stage heart failure (left ventricular assist device and total artificial heart).

### 2.2. Study Participants

To treat aortic stenosis at the Institute, the mechanical prostheses have been used since 1980s. During 2000s, biological valves (conventional, stented bioprostheses) have slowly started to replace mechanical ones for older patients (>65 years). After the new kind of bioprostheses, sutureless technology became commercially available in high-income countries in Europe. The first series of 5 sutureless valves (Perceval) were implanted in 2017 at the Institute. Following this, in 2018, sutureless bioprostheses became routinely used at our institution. In general, sutureless bioprostheses are more convenient in treatment of people who have comorbidities and a small aortic annulus relative to their body surface area.

All people with a severe isolated aortic stenosis who were treated with the Perceval valve at the Institute to date were included in this study. This sample included the first 5 people treated in 2017 and all other patients until 1 July 2021. In line with the study aim, a sample of people treated with the conventional stented bioprostheses (St Jude Epic, St Jude Trifecta and Sorin Crown) at the Institute in the same period (from 1 January 2018 to 1 July 2021) was also included in this analysis.

The inclusion criteria were as follows: having symptomatic isolated aortic stenosis (aortic valve area—AVA ≤ 1 cm^2^ or aortic area index < 0.6 cm^2^/m^2^, having mean gradient of > 40 mmHg and peak velocity of > 4 m/s) and age > 65 years. The exclusion criteria were as follows: having severe aortic regurgitation and active endocarditis, myocarditis or sepsis, having concomitant procedures along with the aortic stenosis, such as the coronary artery disease bypass grafting, mitral valve and aortic surgery and vascular procedures. In addition, the exclusion criteria for the Perceval group were having contraindications for the implantation of this valve (i.e., ratio of sinotubular junction and the aortic annulus diameter of > 1.3 on preoperative echocardiography) and allergy to nickel alloy.

Ethics Committee of the Institute for Cardiovascular Diseases “Dedinje” approved this study. Data were extracted from the medical records of the Institute. For this reason, informed consent was waived. All procedures were carried out in accordance with the World Medical Association’s Declaration of Helsinki for human subject research.

### 2.3. Data Collection

Data were extracted from the official medical records. These included gender, age, anthropometric measurements (height, weight) and body mass index (BMI, calculated as the ratio of weight in kilograms and square height in meters).

The preoperative data comprised the presence of diabetes mellitus, hypertension, hyperlipidemia, peripheral vascular disease, chronic kidney disease, prior stroke, smoking, New York Heart Association (NYHA) classification, aortic stenosis symptoms (angina, dyspnea, syncope), left ventricle ejection fraction (%), AVA (cm^2^), preoperative European System for Cardiac Operative Risk Evaluation (EuroScore) II. The EuroScore II is being used as a predictor of all-cause mortality after major cardiac surgery. It combines an individual’s age, gender, presence of diabetes, chronic pulmonary function, neurological/musculoskeletal dysfunction which affects mobility, renal dysfunction, NYHA class, Canadian Cardiovascular Society (CSS) angina grade, extracardiac arteriopathy, pervious cardiac surgeries, active endocarditis, left ventricle function, recent myocardial infarction, pulmonary artery systolic pressure, urgency, extent of the intervention and thoracic aorta surgery [15].

The operative data included the type of surgical approach (sternotomy, mini-sternotomy and thoracotomy), the type of bioprosthesis (St Jude Epic, St Jude Trifecta, Sorin Crown and Sorin Perceval), operative times (cardiopulmonary bypass time and aortic cross-clamp time). The postoperative data consisted of mechanical ventilation time (in hours), length of stay in the intensive care unit (ICU) (in days), number of blood transfusion units used and re-exploration for bleeding (hemostasis revision).

Postoperative events included wound infection, significant pleural and pericardial effusion needing drainage, postoperative stroke (during hospital stay and during follow-up), new-onset atrial fibrillation, new onset of bundle branch block, permanent pacemaker implantation (during hospital stay and during follow-up), in-hospital and 30-day mortality. Over the follow-up, these outcomes were recorded: all-cause deaths, cardiac-related deaths, stroke, prosthesis endocarditis and needing valve reoperation.

Based on the collected data and few observations, we derived a variable “valve-related complications” which included one or more of the following events: needing permanent pacemaker implantation, having infective endocarditis of the prosthesis and needing valve reoperation.

### 2.4. Follow-up

Mid-term outcomes were retrieved from the medical records, because during the follow-up, the participants were coming to the Institute for their regular clinical and echocardiography checks. For those participants who did not present at the scheduled checks, data about their vital status were retrieved from the records of the Institute of Public Health of Serbia. Outcomes were analyzed in the period August 2021–February 2022. Vital status was ascertained for all participants.

### 2.5. Data Analysis

For the purpose of the study sample description, count and percentage were used for categorical variables and mean/median with corresponding standard deviation/interquartile range were used for continuous variables (based on the normality of the distribution). The differences in categorical variables were assessed by using the chi-square test (for variables with 2 categories) and the chi-square linear-by-linear association (for variables with > 2 categories). The differences in the continuous variables were assessed by using the *t*-test and the Mann Whitney test.

The propensity score analysis was performed. First, a propensity score for valve type allocation (sutureless vs. stented) was created based on age, gender, BMI and surgical approach (sternotomy vs. minimally invasive approach). Then, the propensity score and valve type were analyzed as the independent predictors of all-cause mortality. The choice of the valve (sutureless vs. stented) is mainly dictated by gender, BMI and surgical procedure (sternotomy vs. minimally invasive approach). Moreover, older people are typically treated using a minimally invasive technique to minimize potential surgical trauma and promote faster recovery. For example, women who are overweight and/or obese and women who have a small aortic annulus, in our experience, are generally treated with the sutureless valves. In addition, sutureless valves are placed more often using mini-sternotomy or thoracotomy. For this reason, age, gender, BMI and surgical approach are the key elements to receive a specific treatment (i.e., the type of valve).

Survival of study participants was examined by the Kaplan–Meier curve and logrank test was used to assess the difference in survival between people who received the conventional and the Perceval valve. To assess the predictors of all-cause mortality, the Cox proportional hazard model was used. The dependent variable in the model was the vital status (living—0, died—1). The time included the follow-up time in months for each participant. The independent variables included all the collected demographic, clinical and surgical characteristics as well as the postoperative events. All the independent variables were first tested in a univariate Cox proportional hazard model. All variables univariately associated with the vital status were included in the multivariate Cox proportional hazard model.

The level of statistical significance was at *p* < 0.05. Data analysis was performed using the Statistical Package for Social Sciences 20.0.

## 3. Results

### 3.1. Characteristics of the Study Sample

The study sample comprised a total of 238 people with stented bioprostheses and 101 people with sutureless bioprostheses—Perceval. The characteristics of the study participants according to the valve type are presented in Table 1. Compared to the conventional bioprostheses, the Perceval valve was implanted more often in women. Presence of peripheral vascular disease was more often observed in people who received the Perceval valve. Participants who received the conventional valve more often presented with dyspnea, while syncope were more often present in participants who received the Perceval valve. The mean value of the left ventricle ejection fraction was higher in people with the Perceval valve compared to people who received the conventional valve (Table 1).

### 3.2. Operative Characteristics

Minimally invasive surgical approach (mini-sternotomy and right anterior thoracotomy) was more often applied in people who received the Perceval valve, while full sternotomy was more often applied in people who received the conventional valve (Table 2). Intraoperative and perioperative characteristics are presented in Table 2. The implantation of the Perceval valve was shorter in terms of the cardiopulmonary bypass and aortic cross-clamp time (Table 2).

### 3.3. Perioperative Characteristics and Outcomes of a Median of a 2-Year Follow-up

People who received the Perceval valve also received more transfusion units compared to people who received the conventional valve (Table 3). Table 3 shows perioperative events and outcomes of valve implantation over a median of 2 years of follow-up. The frequency of pleural effusions needing drainage was higher in people who received the Perceval valve. This group of study participants more frequently developed a new onset of bundle branch block after the valve implantation, but there was no difference in terms of the rate of permanent pacemaker implantation over the follow-up.

### 3.4. Propensity Score Analysis

After weighing of the predictive model for propensity scoring, it was observed that the difference in valve allocation was not associated with all-cause mortality (*p* = 0.343) over a median of 2 years of follow-up.

### 3.5. Survival after a Median of 2 Years of Follow-Up

Participants were followed for a median of 2 years (the min–max range of the follow-up was 6–45 months). In that time, 33 out of 238 people who received the conventional valve died (13.9%), and 11 out of 101 people who received the Perceval valve died (10.9%). No difference in the frequency of deaths (all-cause mortality and cardiac mortality) between the groups was observed (Table 3). No difference in survival was observed regarding all-cause mortality and cardiac mortality (Figure 1). Similarly, no difference in the overall survival between the sutureless and stented bioprostheses was found in the subsample of men (*p* = 0.128) and in the subsample women (*p* = 0.155). In addition, no difference in the onset of postoperative strokes, rates of bacterial endocarditis and valve reoperation was observed (Figure 1).

### 3.6. Predictors of All-Cause Mortality in the Total Sample

Because there were no differences in terms of survival between people who received the conventional and the Perceval valve, predictors of all-cause mortality were observed in the total sample of participants. The following characteristics were tested in the univariate Cox regression mode: age, gender, BMI, diabetes mellitus, hypertension, hyperlipidemia, peripheral vascular disease, chronic kidney disease, smoking status, NYHA class, symptoms of aortic stenosis such as angina, dyspnea, syncope, left ventricular ejection fraction, AVA before operation, EuroScore II, surgical approach such as sternotomy, mini-sternotomy and thoracotomy, cardiopulmonary bypass time, aortic cross-clamp time, mechanical ventilation time, length of ICU stay, number of transfusion units, new-onset atrial fibrillation, new onset of bundle branch block, onset of a postoperative stroke and valve-related complications (combined rate of permanent pacemaker implantation, rate of bacterial endocarditis and rate of valve reoperation) (Table 4).

Of all characteristics, older age, higher NYHA class, higher EuroScore II, longer ventilation time, longer ICU stay, receiving blood transfusion, having a stroke over the follow-up and having valve-related complications were univariately associated with all-cause mortality over a median of 2 years of follow-up. These variables entered the multivariate Cox regression model. This model showed that being older (hazard ratio (HR) 1.08; 95% confidence interval 1.02–1.15, *p* = 0.005), having a higher preoperative EuroScore II (HR 1.07; 95% CI 1.03–1.12, *p* = 0.001), having a stroke over the follow-up (HR 16.2, 95% CI 6.76–38.84, *p* = 0.001) and having valve-related complications (HR 2.04, 95%CI 1.04–3.97, *p* = 0.036) were independently associated with dying after the valve implantation over a median of 2 years of follow-up (Table 4).

## 4. Discussion

This study found that people treated for an isolated severe aortic stenosis with stented and sutureless (Perceval) bioprostheses have a similar survival and a similar rate of postoperative complications after a median of 2 years of follow-up. The survival after the implantation of the prostheses was similar in the group of men and in the group of women. In line with previous data [7,8,11], the cardiopulmonary bypass and aortic cross-clamp time was significantly shorter when implanting sutureless vs. stented bioprostheses. A total of 13.9% of people who received the stented and 10.9% of people who received the sutureless bioprostheses died during the follow-up. People who were older, had a higher preoperative EuroScore II, had a postoperative stroke and valve-related complications over the follow-up were more likely to have poor outcomes.

This study is the first to compare the overall survival of people who received the stented and sutureless bioprostheses in Serbia and one of the few available studies conducted in middle-income countries. A study in Turkey (also a middle-income country) explored postoperative outcomes after the implantation of the stented and sutureless valves in a small sample of 52 patients and found no appreciable differences in morbidity and mortality [16]. Our data also showed that Perceval bioprostheses favors minimally invasive surgery, mini-sternotomy and thoracotomy compared to the stented bioprostheses implantation. For this reason, it would be expected that, after the implantation, people who received the sutureless bioprostheses have a lower rate of bleeding and no need for blood transfusion. Nevertheless, this group of patients in our study received more blood transfusion units, while having a similar rate of re-explorations for bleeding compared with the outcomes after the stented bioprostheses implantation. This could potentially be explained by a drop of platelet count as observed in previous studies, albeit without the need for blood transfusion [17,18]. Therefore, our results in terms of needing blood transfusions contrasts the findings from high-income countries [8]. Still, it is not clear why patients with the sutureless valves required more blood transfusion units, and this issue merits further exploration to identify potential underlying causes.

The Perceval valve has been routinely used for more than a decade in high-income countries. Still, there are few prospective cohort studies evaluating its long-term outcomes. One study in Belgium included a maximum follow-up of 11 years, although the median follow-up accounted for 3 years [19]. All-cause mortality after 2 years of follow-up was slightly higher compared to our cohort (13.2% vs. 10.9%) [19], which could be explained by the difference in EuroScore II in the Belgian and our cohort (5.1 vs. 2.3). The EuroScore II was one of the independent predictors of all-cause mortality in our study. This score is a summary measure of health status prior to operation and has been used for the prognosis and prediction of poorer outcomes despite an uneventful surgery. Even a lower EuroScore II, as observed in our study, has shown a predictive value of all-cause mortality in our cohort.

As per the guidelines [4], patients who have the EuroScore II above 8 are recommended a transaortic valve implantation (TAVI) as opposed to the surgical aortic valve replacement (SAVR). People who have the EuroScore II from 4 to 8 may receive either SAVR or TAVI [4]. Echocardiography assessment of this group of patients after either SAVR or TAVI is crucial in terms of reverse myocardial remodeling, valve-related complications, such as bacterial endocarditis, aortic valve regurgitation or paravalvular leakage as well as mitral valve regurgitation, all of which influence the rate of survival [20]. These data could be used as evidence in the decision-making process for better treatment options of severe aortic stenosis. In our cohort of patients, a total of 10 people were assigned the EuroScore II above 8 and were the candidates for TAVI. However, the TAVI procedure in Serbia was unfortunately limited to people who were able to bear the high costs of the intervention out of pocket, as it was not covered by health insurance, until the end of 2021 (i.e., when the follow-up period of our participants was coming to an end).

Strong predictors of mortality after bioprosthesis implantation are the onset of a stroke and valvular complications, such as the endocarditis, pacemaker implantation and needing valve reoperations. Valve reoperations in our study were performed because of the deterioration of the valve due to bacterial endocarditis. These events are, by and large, difficult to eliminate completely and result from numerous intrinsic and individual factors that predispose patients to major adverse events after surgery in general. Some of the drawbacks of the Perceval valve implantation refer to the occurrence of paravalvular leakage, conduction disorders and needing valve reoperations [21]. Evidence suggests that a small proportion of patients (2.3%) experience paravalvular leakage after the Perceval valve implantation [21]. In this study, we did not observe paravalvular leakage, except in one individual (1%). In our cohort, a new onset of bundle branch block was more common after the Perceval valve implantation; however, the permanent pacemaker implantation rate did not correspond to this distribution. A possible reason for such a small rate of pacemaker implantation could be related to the Perceval valve technique implantation [22].

Our experience with the Perceval valve over the past 5 years suggests that it is quite useful and prevents many potential complications during and after aortic valve replacement. The Perceval valve is more convenient in efforts to decrease manipulation of the aortic root and the ascending aorta especially when they are calcified. In this way, it is less likely to disrupt the calcifications and prevent perioperative stroke. In case of a small aortic annulus relative to the body surface area, the Perceval valve is more convenient than the stented bioprosthesis, because of a smoother access to the annulus and the implantation of a prosthesis with a greater effective orifice area. This is especially important when replacing the valves in women who have thin and fragile aortic tissue. In addition, in our experience, people who have concomitant chronic illnesses are already at a higher risk of poorer postoperative outcomes. For this reason, the Perceval valve provides a shorter operative time and a minimal tissue invasion, which is essential to reduce the tissue damage and ensure a better postoperative recovery.

Some problems with the Perceval valve need to be considered. Although the paravalvular leakage is a rare complication after the implantation of the Perceval valve, it could be a serious problem. It may arise as a result of an inadequate anchoring of the valve onto the aortic annulus because of an insufficient decalcification of the annulus or implantation after the bicuspid aortic valve excision. When a major paravalvular leakage is diagnosed using the intraoperative echocardiography right after the implantation of the Perceval valve, it is a major challenge to reposition the valve and, after that, recover the hypertrophic heart from the second cardiac arrest. Despite these limitations, the use of a more costly Perceval valve has considerable advantages for the above mentioned subgroups of patients.

Structural valve deterioration after a long-term follow-up remains yet to be evaluated. Overall, in the forthcoming years, it is essential to prove the durability of the Perceval valve because it seems that it has desirable hemodynamic properties. Durability concerns of the bioprosthesis are essential, and a follow-up of 2 years is not enough to evaluate the safety. Thus, prospective cohort studies are crucial in the assessment of long-term Perceval valve performance [11].

This study has some limitations. Although we have collected data from all patients who were surgically treated at our Institute, the sample size is smaller in comparison to other multicentric studies conducted in high-income countries. This research was retrospective and non-randomized, which is inherently open to selection bias and confounding by indication, although we made effort to minimize this by using the propensity score weighing. A longer follow-up period is needed to accurately assess the durability and hemodynamic performance of the Perceval valve.

## 5. Conclusions

In conclusion, this research conducted in a middle-income country supports previous findings reported in high-income countries about a comparable survival of people who received stented and sutureless bioprostheses. Predictors of all-cause mortality over a median of 2 years of follow-up were older age, having a higher preoperative EuroScore II, having a stroke over the follow-up and having valve-related complications. Survival after bioprostheses implantation should be monitored long-term to ensure optimum postoperative outcomes.

## Figures and Tables

**Figure 1 medicina-59-01032-f001:**
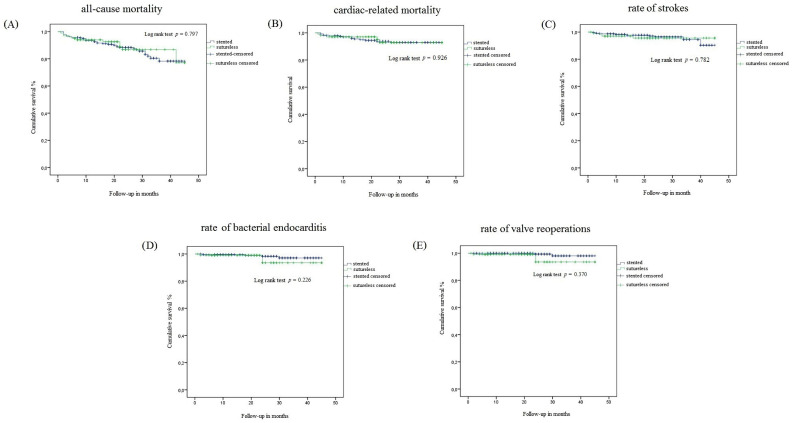
Cumulative survival of people who received the conventional (stented) and the sutureless (Perceval) bioprosthesis in terms of (**A**) all-cause mortality, (**B**) cardiac mortality, (**C**) rate of postoperative strokes, (**D**) bacterial endocarditis and (**E**) valve reoperation.

**Table 1 medicina-59-01032-t001:** Characteristics of patients before aortic valve replacement according to valve prosthesis.

Variables		Valve Type		*p*
		Stented Bioprosthesis N = 238 *n* (%)	Sutureless Perceval N = 101 *n* (%)	
Age (mean ± SD)		72.8 ± 5.3	72.1 ± 5.6	0.360
Gender	men	150 (63.0)	33 (32.7)	<0.001
	women	88 (37.0)	68 (67.3)	
Body mass index		27.8 ± 4.1	27.4 ± 4.9	0.412
Diabetes mellitus		60 (25.2)	27 (26.7)	0.769
Hypertension		221 (92.9)	89 (88.1)	0.154
Hyperlipidemia		143 (60.1)	71 (70.3)	0.075
Peripheral vascular disease		30 (12.6)	25 (24.8)	0.006
Chronic kidney disease		25 (10.5)	8 (7.9)	0.463
Prior stroke		14 (5.9)	10 (9.9)	0.187
Smoking		109 (45.8)	42 (41.6)	0.475
NYHA	I	22 (9.2)	6 (5.9)	0.700
	II	141 (59.2)	59 (58.4)	
	III	72 (30.3)	35 (34.7)	
	IV	3 (1.3)	1 (1)	
Aortic stenosis symptoms	angina	89 (37.4)	29 (28.7)	0.125
	dyspnea	196 (82.4)	23 (22.8)	<0.001
	syncope	41 (17.2)	40 (39.6)	<0.001
LV ejection fraction (%)		50.6 ± 11.4	53.3 ± 9.0	0.032
AVA before operation (cm^2^)		0.7 ± 0.2	0.7 ± 0.2	0.670
Preoperative EuroScore II (mean ± SD)		2.55 ± 3.86	2.32 ± 1.74	0.560

Legend: NYHA—Ney York Heart Association classification; LV—left ventricle; AVA—aortic valve area.

**Table 2 medicina-59-01032-t002:** Operative and perioperative characteristics according to valve prosthesis.

Variables		Valve Type		*p*
		Stented Bioprosthesis N = 238 *n* (%)	Sutureless Perceval N = 101 *n* (%)	
Surgical approach	Sternotomy	159 (66.8)	14 (13.9)	<0.001
	Mini-sternotomy	77 (32.4)	44 (43.6)	0.049
	Thoracotomy	2 (0.8)	43 (42.6)	<0.001
Aortic valve prosthesis	St Jude Epic	183 (76.9)	n/a	n/a
	St Jude Trifecta	45 (19.9)	n/a	
	Sorin Crown	10 (4.2)	n/a	
	Perceval	n/a	101 (100.0)	
Cardiopulmonary bypass time in minutes (mean ± SD)		89.5 ± 25.2	83.0 ± 28.9	0.038
Aortic cross-clamp time in minutes (mean ± SD)		63.7± 19.3	49.5 ± 14.4	<0.001
Mechanical ventilation time in hours		17.0 ± 7.0	16.0 ± 7.0	0.240
ICU stay in days		2.0 ± 1.0	2.0 ± 2.0	0.395

Legend: n/a—not applicable; ICU—intensive care unit.

**Table 3 medicina-59-01032-t003:** Perioperative events and outcomes after a median of 2 years of follow-up.

Variable		Valve Type		*p*
		Stented Bioprosthesis N = 238 *n* (%)	Sutureless Perceval N = 101 *n* (%)	
Transfusion units	0	143 (60.1)	42 (41.6)	0.002
	≥1	95 (39.9)	59 (58.4)	
Hemostasis revision		16 (6.7)	9 (8.9)	0.481
Wound infection		4 (1.7)	2 (2.0)	0.848
Pleural effusion		13 (5.5)	13 (12.9)	0.019
Pericardial effusion		9 (3.8)	3 (3.0)	0.712
Intrahospital stroke		3 (1.3)	1 (1.0)	0.833
New-onset atrial fibrillation		58 (24.4)	16 (15.8)	0.082
New onset of bundle branch block		11 (4.6)	18 (17.8)	<0.001
Permanent pacemaker implantation (intrahospital and follow-up)		4 (1.7)	3 (3.0)	0.445
Intrahospital/30-day mortality		1 (0.4)	2 (2.0)	0.161
Major adverse cardiac and cardiovascular events after a median of 2 years	Myocardial infarction	2 (0.8)	0 (0)	0.355
	Fatal stroke	6 (2.5)	4 (4.0)	0.473
	Non-fatal stroke	3 (1.3)	0 (0)	0.257
Total deaths after a median of 2 years		33 (13.9)	11 (10.9)	0.400
Cardiac deaths after a median of 2 years ^†^		14 (5.9)	5 (5.0)	0.733
Non-cardiac deaths after a median of 2 years ^‡^		29 (12.2)	6 (5.9)	0.104
Valve complications after a median of 2 years	Bacterial endocarditis	4 (1.7)	3 (3.0)	0.445
	Rate of valve reoperation	2 (0.8)	3 (3.0)	0.137
Total valve-related complications *		17 (7.1)	8 (7.9)	0.270

Legend: * permanent pacemaker implantation, bacterial endocarditis of prosthesis and valve reoperation; ^†^ myocardial infarction, sudden cardiac death, malignant arrhythmia, heart failure; ^‡^ malignant tumors (colon, kidney, omentum, prostate, liver, stomach, ovary), traumatic injury (falling, car accident), acute kidney failure.

**Table 4 medicina-59-01032-t004:** Factors associated with all-cause mortality over a median of 2 years of follow-up.

Variable	HR	95% CI	*p*	HR	95% CI	*p*
Age	3.60	1.89–6.89	<0.001	1.08	1.02–1.15	0.005
Gender	0.72	0.39–1.31	0.286			
Stented vs. sutureless	0.82	0.41–1.65	0.592			
Body mass index	0.99	0.92–1.06	0.875			
Diabetes mellitus	1.67	0.89–3.11	0.105			
Hypertension	0.83	0.32–2.14	0.712			
Hyperlipidemia	1.12	0.61–2.08	0.699			
Peripheral vascular disease	0.92	0.42–2.00	0.846			
Chronic kidney disease	1.67	0.70–3.96	0.240			
Prior stroke	1.12	0.40–3.16	0.817			
Smoking	0.61	0.32–1.13	0.120			
NYHA class	1.99	1.22–3.24	0.006	1.29	0.75–2.22	0.354
Angina	0.78	0.41–1.47	0.444			
Dyspnea	1.42	0.73–2.75	0.297			
Syncope	0.57	0.26–1.26	0.171			
Left ventricle ejection fraction	1.00	0.97–1.03	0.787			
AVA before operation	1.23	0.25–5.94	0.798			
Preoperative EuroScore II	3.55	1.82–6.93	<0.001	1.07	1.03–1.12	0.001
Sternotomy	1.56	0.83–2.91	0.162			
Mini-sternotomy	0.72	0.37–1.41	0.346			
Thoracotomy	0.61	0.18–1.99	0.417			
Cadio-pulmonary bypass time	1.00	0.99–1.01	0.129			
Aortic cross-clamp time	0.99	0.98–1.01	0.776			
Mechanical ventilation time	3.24	1.51–6.97	0.003	1.00	0.99–1.00	0.488
ICU stay	1.02	1.01–1.03	<0.001	0.94	0.86–1.03	0.202
Transfusion units	1.11	1.06–1.17	<0.001	1.20	0.95–1.50	0.112
New-onset atrial fibrillation	1.50	0.80–2.84	0.208			
New onset of bundle branch block	0.84	0.26–2.71	0.769			
Total stroke after a median of 2 years	11.63	5.70–23.72	<0.001	16.20	6.76–38.84	0.001
Valve-related complications during follow-up *	2.59	1.77–3.79	<0.001	2.04	1.04–3.97	0.036

Legend: HR—hazard ratio; CI—confidence interval; NYHA—New York Heart Association classification; AVA—aortic valve area; ICU—intensive care unit; * permanent pacemaker implantation, infective endocarditis of prosthesis and valve reoperation.

## Data Availability

The dataset underlying this study is available upon a reasonable request from the corresponding author.

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
