# Peer review of "An Analysis of Early Results after Valve Replacement in Isolated Aortic Valve Stenosis by Using Sutureless vs. Stented Bioprostheses: A Single-Center Middle-Income Country Experience"

_medicina, 2023, doi:10.3390/medicina59061032_

Round 1

Reviewer 1 Report

The authors present their experience with 339 patients having isolated aortic stenosis and treated with  valve replacement by a stented or a sutureless prosthesis   between 2018 and 2021. Aim  of the study  was to compare at 2 years median follow-up time the overall survival of the patients implanted with each of the two bioprostheses.

The topic of the study is newsworthy , the manuscript is well written and the statistical analysis is appropriate . The evaluation of predictors of all-cause mortality in the two groups  by univariate and multivariate analysis  is accurate and interesting.

However, this paper presents , in my opinion, some major drawbacks.

·         Due to the  presence of several confounding factors as the various type of stented prostheses used ( Epic. Trifecta, Crown), the different surgical techniques chosen ( conventional or minimal invasive  approach ) and the different size of the two cohorts along with their in-homogeneity, a propensity-matched analysis would have been more appropriate.

·         The authors relied on the sole variable of patients overall mortality in order  to assert their conclusions . Indeed,  the causes of death for each patients were not reported at all. Therefore the deaths could have been related  to cardiac or non-cardiac reasons. This is a crucial drawback of this study.

·         The table 4  is insubstantial with a comparison  study  and should be omitted.

·         The basic variables necessary  to assess  the results  after implantation of different types of cardiac prostheses are, besides overall mortality , cardiac mortality, incidence of bacterial endocarditis and MACCE, structural valve deterioration, rate of reoperation. All these parameters should be taken in account to achieve a correct evaluation of any aortic bio-prosthesis outcome, and should be illustrated preferably  by using KM curves.

·         In Figure 1 the overall survival KM  should be cut at 45 months follow-up time

·         Anyway , 2 years follow-up is a too short timeframe to evaluate the outcome after  a bio-prosthetic aortic valve replacement . The majority of biological valve substitutes currently  in use  perform quite well up to 10 years after implantation. Therefore a study comparing different bio-prostheses and reporting outcomes at shorter than that follow-up times are of scarce or no  interest to the surgical audience.

·         I would suggest the authors  to change the title of this paper as follows:   ' An analysis of early   results after valve replacement in isolated aortic valve stenosis by using  Sutureless vs. Stented bioprostheses. A single-center , middle income country experience ' and to re-write the manuscript accordingly.

Author Response

Reviewer 1

The authors present their experience with 339 patients having isolated aortic stenosis and treated with valve replacement by a stented or a sutureless prosthesis between 2018 and 2021. The aim of the study was to compare at 2 years of median follow-up time the overall survival of the patients implanted with each of the two bioprostheses.

The topic of the study is newsworthy, the manuscript is well-written and the statistical analysis is appropriate. The evaluation of predictors of all-cause mortality in the two groups by univariate and multivariate analysis is accurate and interesting. However, this paper presents, in my opinion, some major drawbacks.

Due to the presence of several confounding factors as the various type of stented prostheses used (Epic, Trifecta, Crown), the different surgical techniques chosen (conventional or minimal invasive approach) and the different size of the two cohorts along with their in-homogeneity, a propensity-matched analysis would have been more appropriate.

ANSWER 1: We agree. The propensity score analysis has now been performed. First, a propensity score for valve type allocation (sutureless vs. stented) was created based on age, gender, BMI and surgical approach (sternotomy vs. minimally invasive approach). Then, the propensity score and valve type were analyzed as the independent predictors of all-cause mortality.

After weighing of the predictive model for propensity scoring, it was observed that difference in valve allocation was not associated with all-cause mortality (p=0.343) over a median of 2 years of follow-up.

We have now added this in the manuscript.

The authors relied on the sole variable of patients' overall mortality in order to assert their conclusions. Indeed, the causes of death for each patients were not reported at all. Therefore the deaths could have been related to cardiac or non-cardiac reasons. This is a crucial drawback of this study.

ANSWER 2: We agree. We have now added the rates of cardiac and non-cardiac deaths according to valve type and explained in the footnote causes of death. Additionally, we have now performed the Kaplan-Meier analysis for both all-cause mortality and cardiac mortality.

The table 4  is insubstantial with a comparison  study  and should be omitted.

ANSWER 3: We agree. We have now omitted Table 4. The HR estimates are presented in the body of the manuscript.

The basic variables necessary to assess the results after implantation of different types of cardiac prostheses are, besides overall mortality, cardiac mortality, incidence of bacterial endocarditis and MACCE, structural valve deterioration, rate of reoperation. All these parameters should be taken in account to achieve a correct evaluation of any aortic bio-prosthesis outcome, and should be illustrated preferably by using KM curves.

ANSWER 4: We agree. We have now included the rates of cardiac mortality, incidence of bacterial endocarditis and MACCE (myocardial infarction, fatal and non-fatal stroke) and rate of reoperation. All people who had valve reoperation also had a structural valve deterioration due to endocarditis. For this reason, we did not include the variable "structural valve deterioration" in Table 3. We have explained this in the manuscript.

We have also illustrated the KM curves for cardiac mortality, total stroke, bacterial endocarditis and valve reoperation.

In Figure 1 the overall survival KM  should be cut at 45 months follow-up time.

ANSWER 5: We agree. We have now cut the follow-up time to 45 months.

Anyway, 2 years follow-up is a too short timeframe to evaluate the outcome after a bio-prosthetic aortic valve replacement . The majority of biological valve substitutes currently  in use  perform quite well up to 10 years after implantation. Therefore a study comparing different bio-prostheses and reporting outcomes at shorter than that follow-up times are of scarce or no  interest to the surgical audience.

ANSWER 6: We agree that this follow-up time is relatievly short, however, Serbia is a middle income coutry and has limited availability of sutureless prostheses. These 101 patients who were implanted sutureless valves are the first sutureless valves implanted in Serbia. For this reason, these results represent our first 5-year experience ever with sutureless prostheses. Furthermore, these single-center analyses have wider scientific implications because of potential meta-analyses, that should include all available data worldwide.  

I would suggest the authors  to change the title of this paper as follows: 'An analysis of early   results after valve replacement in isolated aortic valve stenosis by using  Sutureless vs. Stented bioprostheses. A single-center , middle income country experience' and to re-write the manuscript accordingly.

ANSWER 7: We agree. We have now modified the manuscript title as suggested.

Reviewer 2 Report

Marko Kaitovic et al presented the manuscript "PERCEVAL VS. CONVENTIONAL VALVE IMPLANTATION IN TREATMENT OF AN ISOLATED SEVERE AORTIC STENOSIS: A COMPARATIVE SURVIVAL ANALYSIS FROM A SINGLE-CENTER IN A MIDDLE-INCOME COUNTRY". An observational study comparing suturless vs stented bioprothesis in aortic stenosis patients.

Some comments:

1) In the introduction, the Authors said surgical aortic valve replacement represents the gold standard for treatment of severe aortic stenosis. I would suggest to further discuss the point comparing the overall data among surgical vs TAVR.

2) Do the Authors have baseline characteristics info? This is very important to understand differences among study groups considering the observational nature of the study. The same for surgery details. I'm not able to see the the table 1 and table 2.

3) About outcomes. I suggest to further specify if the Authors assessed adverse events in a single composite outcome or not. This is behind the "valve-related complications". The table 3 is not available.

4) The Authors have to be more specific about the variables included in the predict model of all-cause mortality. The table 4 is not available.

5) I would suggest to add HR and 95% CI along the figure 1

To be accurate in this revision, it's important to have access to 100% of the data.

Author Response

Reviewer 2

Marko Kaitovic et al presented the manuscript "PERCEVAL VS. CONVENTIONAL VALVE IMPLANTATION IN TREATMENT OF AN ISOLATED SEVERE AORTIC STENOSIS: A COMPARATIVE SURVIVAL ANALYSIS FROM A SINGLE-CENTER IN A MIDDLE-INCOME COUNTRY". An observational study comparing suturless vs stented bioprothesis in aortic stenosis patients.

Some comments:

1) In the introduction, the Authors said surgical aortic valve replacement represents the gold standard for treatment of severe aortic stenosis. I would suggest to further discuss the point comparing the overall data among surgical vs TAVR.

ANSWER 1: We agree. We have now modified the sentence to read: "Surgical aortic valve replacement is still a standard treatment of low-risk patients with severe aortic stenosis in order to preserve life expectancy [3, 4]."

In the Discussion, we have further discussed the SAVR vs. TAVR.

Discussion, paragraph 4:

"... As per the guidelines [4], patients who have EuroScore II above 8 are recommended a transaortic valve replacement (TAVR) as opposed to the surgical valve replacement. In our cohort of patients, a total of 10 people were assigned an EuroScore II above 8 and were the candidates for TAVR. However, the TAVR procedure in Serbia was unfortunately limited to people who were able to bear the costs of the intervention out of pocket, as it was not covered by health insurance, until 2021 (i.e. when the follow-up period of our participants was coming to an end)."

2) Do the Authors have baseline characteristics info? This is very important to understand differences among study groups considering the observational nature of the study. The same for surgery details. I'm not able to see the the table 1 and table 2.

ANSWER 2: Indeed, all baseline characteristics info is presented in tables 1-3. We do not know why the tables were not visible, as all tables were submitted. We hope that now in the revised version all tables visible.

3) About outcomes. I suggest to further specify if the Authors assessed adverse events in a single composite outcome or not. This is behind the "valve-related complications". The table 3 is not available.

ANSWER 3: We have now specified that the observed outcomes in the Kaplan Meier curve were all-cause mortality, cardiac mortality, rate of postoperative strokes, rate of bacterial endocarditis and rate of valve reoperation.

The variable "valve related complications" comprised permanent pacemaker implantation, bacterial endocarditis and valve reoperation, as these three single variables had few observations on their own. If single variable with few observations were tested in the model, they would decrease the statistical power.

4) The Authors have to be more specific about the variables included in the predict model of all-cause mortality. The table 4 is not available.

ANSWER 4: We agree. We have now clarified all the variables that were univariately tested in the Cox proportional hazard model. As per the comment of Reviewer 1 (answer no. 3), we were suggested to remove Table 4 from the results.

Results, Predictors of all-cause mortality in the total sample

"... The following characteristics were tested in the univariate Cox regression mode: age, gender, body mass index, diabetes mellitus, hypertension, hyperlipideia, peripheral vascular disease, chronic kidney disease, smoking status, NYHA class, symptoms of aortic stenosis such as angina, dyspnea, synkope, left ventricular ejection fraction, AVA before operation, EuroScore II, surgical approach such as sternotomy, mini-sternotomy and thoracotomy, cardiopulmonary bypass time, aortic cross-clamp time, mechanical ventilation time, length of ICU stay, number of transfusion units, de novo atrial fibrillation, de novo heart block, onset of postoperative stroke and valve-related complications (combined permanent pacemaker implantation, rate of bacterial endocarditis and rate of valve reoperation)."

5) I would suggest to add HR and 95% CI along the figure 1

ANSWER 6: We agree. We have included the p value in each part of Figure 1.

To be accurate in this revision, it's important to have access to 100% of the data.

ANSWER 7: We agree. All data are presented in Tables 1-3. We hope that all tables are now visible for review.

Reviewer 3 Report

This is a study that aims to evaluate the survival in patients with isolated severe aortic stenosis after the implantation of sutureless- versus stented bioprosthesis surgical valves in middle-income countries. The authors found comparable survival in both groups in high-income countries. And they found the predictors of all-cause mortality over 2 years in those patients were older age, high EuroSCORE II, stroke after surgery, and valve-related complications. It is a very interesting and meaningful article, nevertheless, the authors need to be more clarified the following.

1. Generally speaking, the sutureless valve over stented bioprosthetic valve has several strengths in reduction of bypass time, less manipulation of the calcific aorta, and then it has more benefits in sicker patients. However, in this study, the authors showed shortened bypass times in sutureless valve implantation, but there were no differences in surgical times, ICU stay, re-do operation, or mortality. And even the acquirement of transfusion was more common in patients with sutureless valve implantation.  

2. In accordance with the above, the authors would be better to suggest as follows,

 1) add the echocardiographic data before- and after surgery to a different table

 2) the authors would be better to remove ‘death, cardiac death, stroke, endocarditis, and re-do operation for 2 years’ and instead there were presented separate Kaplan-Meier graphs in conjunction with Figure 1.

 3) In Figure 1, if possible, the authors would be better to truncate the follow-up duration at 2 years and present the log-rank p valve.

3. If possible, the authors would be better to add the health-economic benefits of the sutureless valve according to compare the total hospital costs for both groups. Therefore, the authors would be able to suggest the benefit of sutureless valves over conventional surgical valves in mid-income countries.

4. In Table 3,4, the authors would be better to specify the ‘de novo heart block’ such as a new-onset of bundle branch block or high-degree block for the reduction of confusion.

5. In Lines 268 to 274, the authors explained the reason for the higher frequency of transfusion units in patients with sutureless valves due to the decrease the platelet counts. Would you please show the data in detailed table 3 and the discussion section? Together, the authors would explain the adequate theory. The reduction of platelet counts after mechanical valves such as TAVR is common, but it is usually transient and does not need transfusion.  

6. It is very important to the durability concerns in bioprosthesis, the 2-year follow-up is too short to evaluate the safety. And then the authors would be mentioned in the limitation. Together, the authors would be better to mention the limitations more specifically including above mentioned issues.

7. Minor comments

 1) line 180, not ‘dispnea’, but ‘dyspnea or dyspnoea’

 2) lines 185 and 243, NYHA indicates New York Heart Association

 3) line 318, not ‘implementation’, but ‘implantation’

Author Response

Reviewer 3

This is a study that aims to evaluate the survival in patients with isolated severe aortic stenosis after the implantation of sutureless- versus stented bioprosthesis surgical valves in middle-income countries. The authors found comparable survival in both groups in high-income countries. And they found the predictors of all-cause mortality over 2 years in those patients were older age, high EuroSCORE II, stroke after surgery, and valve-related complications. It is a very interesting and meaningful article, nevertheless, the authors need to be more clarified the following.

  1. Generally speaking, the sutureless valve over stented bioprosthetic valve has several strengths in reduction of bypass time, less manipulation of the calcific aorta, and then it has more benefits in sicker patients. However, in this study, the authors showed shortened bypass times in sutureless valve implantation, but there were no differences in surgical times, ICU stay, re-do operation, or mortality. And even the acquirement of transfusion was more common in patients with sutureless valve implantation.

ANSWER 1: We agree. We were also pleased to observe that the relevant surgical characteristics did not differ between the two groups of patients. The observed difference in transfusion units was surprising indeed. However, it did not influence the all-cause mortality in the final multivariate prediction model.

  1. In accordance with the above, the authors would be better to suggest as follows,

 1) add the echocardiographic data before- and after surgery to a different table

ANSWER 2.1: We agree that these pieces of information are relevant. However, the complete echocardiographic findings before and after the operation have already been part of another manuscript of our that is currently under review in another journal. Due to the possible overlapping and autoplagiarism, we are compelled to omit these data from the present manuscript to respect research ethics principles.

 2) the authors would be better to remove ‘death, cardiac death, stroke, endocarditis, and re-do operation for 2 years’ and instead there were presented separate Kaplan-Meier graphs in conjunction with Figure 1.

ANSWER 2.2: We agree. We have now presented the Kaplan-Meier curves for cardiac mortality, total stroke, bacterial endocarditis and valve reoperation.

 3) In Figure 1, if possible, the authors would be better to truncate the follow-up duration at 2 years and present the log-rank p valve.

ANSWER 2.3: We agree. However, Reviewer no. 1 suggested that the follow-up time be truncated at 45 months, which we have implemented. We have also included the log rank test p value in each figure, as per the suggestion.

  1. If possible, the authors would be better to add the health-economic benefits of the sutureless valve according to compare the total hospital costs for both groups. Therefore, the authors would be able to suggest the benefit of sutureless valves over conventional surgical valves in mid-income countries.

ANSWER 3: We agree that these pieces of information are relevant. However, the complete health economics analysis has already been part of another manuscript of our that is currently under review in another journal. Due to the possible overlapping and autoplagiarism, we are compelled to omit these data from the present manuscript to respect research ethics principles.

  1. In Table 3,4, the authors would be better to specify the ‘de novo heart block’ such as a new-onset of bundle branch block or high-degree block for the reduction of confusion.

ANSWER 4: We agree. We have now used the term "new-onset of bundle branch block" throughout the manuscript.

  1. In Lines 268 to 274, the authors explained the reason for the higher frequency of transfusion units in patients with sutureless valves due to the decrease the platelet counts. Would you please show the data in detailed table 3 and the discussion section? Together, the authors would explain the adequate theory. The reduction of platelet counts after mechanical valves such as TAVR is common, but it is usually transient and does not need transfusion.

ANSWER 5: We agree and we hope that table 3 is now visible, as it was not previously visible to reviewers. The difference in the overall transfused units was found between patients who received stented and sutureless valves. This could potentially be explained by a drop of platelet count as observed in previous studies, albeit without the need for blood transfusion. Still, it is not clear why patients with sutureless valves required more blood transfusion units and this issue merits futher exploration to identify potential underlying causes. We will explore this issue in detail in our next research. We have now commented this in the Discussion.

  1. It is very important to the durability concerns in bioprosthesis, the 2-year follow-up is too short to evaluate the safety. And then the authors would be mentioned in the limitation. Together, the authors would be better to mention the limitations more specifically including above mentioned issues.

ANSWER 6: We agree. We have now commented on this in the limitations paragraph. And we have also changed the title of the manuscript to acknowledge the mid term outcomes.

  1. Minor comments

 1) line 180, not ‘dispnea’, but ‘dyspnea or dyspnoea’

ANSWER 7.1: We agree. We have now corrected this typo.

 2) lines 185 and 243, NYHA indicates New York Heart Association

 ANSWER 7.2: We agree. We have now corrected this typo.

 3) line 318, not ‘implementation’, but ‘implantation’

ANSWER 7.3: We agree. We have now corrected this typo.

Round 2

Reviewer 2 Report

Marko Kaitovic et al presented the revised version of the manuscript "PERCEVAL VS. CONVENTIONAL VALVE IMPLANTATION IN TREATMENT OF AN ISOLATED SEVERE AORTIC STENOSIS: A COMPARATIVE SURVIVAL ANALYSIS FROM A SINGLE-CENTER IN A MIDDLE-INCOME COUNTRY".

This updated version looks substantially improved and I have only few additional comments:

1) Why the Authors did not include in PSM all variables which resulted significantly different at baseline? This of course could improve the quality of the data.

2) The Authors should explain how they selected the variables included among the Cox model. I would suggest to include the model in a table to help the readers but it is not a major point.

3) Do the Authors have further info about post-procedural echocardiography? In particular for prosthesis function as well as cardiac function changes after TAVI. The Authors should add and discuss this ref according: Angellotti D, et al. Echocardiographic Evaluation after Transcatheter Aortic Valve Implantation: A Comprehensive Review. Life.2023; 13(5):1079.

Author Response

Dear Editor,

            We are pleased to submit our second revised original research entitled “An analysis of early results after valve replacement in isolated aortic valve stenosis by using sutureless vs. stented bioprostheses: a single-center middle income country experience” by Kaitovic et al. for publication in Medicina (Kaunas).

            We are very grateful for the comments of the reviewers. All comments of the reviewers were accepted. We answered in a point-by-point manner to all the issues raised. All changes in the manuscript were highlighted in color red.         

            Please let us know whether our manuscript is found to be suitable for publication in Medicina (Kaunas).

Sincerely,

Tatjana Gazibara and Marko Kaitovic

Reviewer 2

Marko Kaitovic et al presented the revised version of the manuscript "PERCEVAL VS. CONVENTIONAL VALVE IMPLANTATION IN TREATMENT OF AN ISOLATED SEVERE AORTIC STENOSIS: A COMPARATIVE SURVIVAL ANALYSIS FROM A SINGLE-CENTER IN A MIDDLE-INCOME COUNTRY".

This updated version looks substantially improved and I have only few additional comments:

1) Why the Authors did not include in PSM all variables which resulted significantly different at baseline? This of course could improve the quality of the data.

ANSWER 1: We agree that this is an important issue and needs to be clarified further. The choice of the valve (stented vs. sutureless prosthesis) is mainly dictated by gender, body mass index and surgical procedure (strenotomy vs. minimally invasive approach). Also, older people are typically treated using a minimally invasive tehcnique to minimize potential surgical trauma and favorize faster recovery. For example, women who are overweight and/or obese with a small arotic annulus in our hospital are treated with sutureless valves. In addition, sutureless valves are placed more often using mini-sternotomy or thoracotomy. For this reason, age, gender, BMI and surgical approach are the key elements to receive specific treatment (i.e. type of valve).

Other baseline characteristics which showed significant difference between the groups (i.e. peripheral vascular disease and aortic stenosis symptoms) were not the factors based on which the choice of valve is made. We have now explained this in the Manuscript (Methods, Data analysis, paragrapah 2)

2) The Authors should explain how they selected the variables included among the Cox model. I would suggest to include the model in a table to help the readers but it is not a major point.

ANSWER 2: We agree. We have now included Table 4 to provide coefficients from the Cox proportional hazard model to clarify how varibles for the multivariate mode were chosen.

Results, Predictors of all-cause mortality:

The following characteristics were tested in the univariate Cox regression mode: age, gender, body mass index, diabetes mellitus, hypertension, hyperlipidemia, peripheral vascular disease, chronic kidney disease, smoking status, NYHA class, symptoms of aortic stenosis such as angina, dyspnea, synkope, left ventricular ejection fraction, AVA before operation, EuroScore II, surgical approach such as sternotomy, mini-sternotomy and thoracotomy, cardiopulmonary bypass time, aortic cross-clamp time, mechanical ventilation time, length of ICU stay, number of transfusion units, new-onset atrial fibrillation, new-onset of bundle branch block, onset of postoperative stroke and valve-related complications (combined permanent pacemaker implan-tation, rate of bacterial endocarditis and rate of valve reoperation) (Table 4).

            Of all charasteristics, older age, higher NYHA grade, higher EuroScore II, longer ventilation time, longer ICU stay, receiving blood transfusion, having stroke over fol-low-up and having valve-related complications were univariately associated with all-cause mortality over a median of 2 years of follow-up. These variables entered the multivariate Cox regression model. This model showed that being older (hazard ratio [HR] 1.08; 95% confidence interval 1.02-1.15, p=0.005), having a higher preoperative EuroScore II (HR 1.07; 95% CI 1.03-1.12, p=0.001), having stroke over follow-up (HR 16.2, 95% CI 6.76-38.84, p=0.001) and having valve-related complications (HR 2.04, 95%CI 1.04-3.97, p=0.036) were independently associated with dying after valve implantation over a median of 2 years of follow-up (Table 4).

3) Do the Authors have further info about post-procedural echocardiography? In particular for prosthesis function as well as cardiac function changes after TAVI. The Authors should add and discuss this ref according: Angellotti D, et al. Echocardiographic Evaluation after Transcatheter Aortic Valve Implantation: A Comprehensive Review. Life.2023; 13(5):1079.

ANSWER 3: We agree that info about post-procedural echocardiography is relevant. However, the complete echocardiographic findings before and after the operation have already been part of another manuscript of our that is currently under review in another journal. Due to the possible overlapping and autoplagiarism, we are compelled to omit these data from the present manuscript to respect research ethics principles.

We have now discussed the recommended reference as follows (Discussion, paragrpah 4):

As per the guidelines [4], patients who have the EuroScore II above 8 are recommended a transaortic valve implantation (TAVI) as opposed to the surgical aortic valve replacement (SAVR). People who have the EuroScore II from 4 to 8 may receive either SAVR or TAVI [4]. Echocardiography assessment of this group of patients after either SAVR or TAVI is crucial in terms of reverse myocardial remodeling, valve-related complications, such as bacterial endocarditis, aortic valve regurgitation or paravalvular leakage as well as mitral valve regurgitation, all of which influence the rate of survival [20]. These data could be used as evidence in the decision-making process for better treatment options of severe aortic stenosis. In our cohort of patients, a total of 10 people were assigned the EuroScore II above 8 and were the candidates for TAVI. However, the TAVI procedure in Serbia was unfortunately limited to people who were able to bear the high costs of the intervention out of pocket, as it was not covered by health insurance, until the end of 2021 (i.e. when the follow-up period of our participants was coming to an end).

We have also included the reference by Angellotti et al. 2023 in the reference list.
